# Effect of Particle Size of Wheat and Barley Grain on the Digestibility and Fermentation of Carbohydrates in the Small and Large Intestines of Growing Pigs

**DOI:** 10.3390/ani13121986

**Published:** 2023-06-14

**Authors:** Geon-Il Lee, Knud Erik Bach Knudsen, Mette Skou Hedemann

**Affiliations:** Department of Animal and Veterinary Sciences, Faculty of Technical Sciences, Aarhus University, Blichers Allé 20, 8830 Tjele, Denmark; clerk123@naver.com (G.-I.L.); knuderik.bachknudsen@anivet.au.dk (K.E.B.K.)

**Keywords:** cereals, fiber, particle size, pigs, wheat, barley

## Abstract

**Simple Summary:**

The price of feed ingredients has been drastically increasing over the last decade, and thus the methods for improving nutrient digestibility and utilization with various feed ingredients have been widely studied in the field of the swine industry. One of the most efficacious approaches to enhancing nutrient digestibility and utilization is the manipulation of feed particle size. Nonetheless, limited research has been conducted on the impact of varying particle sizes of feeds with high fiber content. Dietary fiber is composed of several components, including β-glucan, arabinoxylan, and cellulose, which exhibit distinct physiological effects on the small and large intestines. The compositions and proportions of these components can influence and modulate nutrient digestibility in the small and large intestines of pigs. The objective of the present investigation was to examine the potential for enhanced nutrient and fiber digestibility in growing pigs through the reduction of particle size in barley and wheat diets. Similar ileal and total nutrient digestibility were observed in pigs fed diets containing fine and coarse wheat; however, pigs consuming coarse barley exhibited reduced nutrient digestibility compared to the other diets. Consequently, the present study shows that nutrient digestibility was more influenced by reduced particle size in barley than wheat, most likely because of the rigid fiber structure of barley and barley hull.

**Abstract:**

The objective of this investigation was to study the effects of different cereal types, barley and wheat, with different particle sizes (PS) on the recovery of ileal digesta and fecal excretion, digestion of nutrients and fiber components, mean transit time (MTT), and short-chain fatty acid content and composition in growing pigs studied in two experiments. Five barrows with ileal cannulas (initial BW 35.9 ± 1.5 kg) in Experiment 1 and thirty-two castrated pigs (30.8 ± 1.3 kg) in Experiment 2 were fed four different diets: barley fine, barley coarse, wheat fine and wheat coarse diets. The cereal type and PS did not influence the relative weight of the small and large intestines and pH of digesta, whereas MTT in the large intestine of pigs fed the coarse barley diet was lower compared to pigs fed other diets (*p* < 0.05). Pigs fed the coarse barley diet had lower apparent ileal digestibility (AID) and apparent total tract digestibility (ATTD) of nutrients and fiber (*p* < 0.05), whereas pigs fed the fine barley diet had similar AID and ATTD to pigs fed wheat fine and coarse diets (*p* < 0.05). In conclusion, the barley diet was more influenced by PS in comparison to wheat, thereby inducing lower AID and ATTD of nutrient.

## 1. Introduction

The pig is a monogastric animal with endogenous enzymes playing a crucial role in its digestive process [1]. It is therefore essential to provide pigs with high-quality feed that provides nutrients in a readily available form for the enzymes to digest. Apart from the composition of the feed, different feed structures and forms, such as particle size (PS), extrusion, pellet, flake and cooking, significantly affect the efficiency of feed and nutrient utilization. Therefore, they should be optimized for nutrient absorption [2]. Ensuring the provision of adequate essential nutrients to meet the nutritional requirements of pigs presents a challenge due to diverse feed processing techniques and ingredient variations that can significantly impact nutrient utilization [3]. In addition, these effects may vary depending on the growth stage of the pigs [4]. However, finely ground materials can have a negative impact, particularly on gastric ulcers [5].

Barley and wheat are quantitatively the most important components of diets for growing pigs in Denmark and most other European countries. Both cereals have a high concentration of carbohydrates, predominantly as polysaccharides, including starch and non-starch polysaccharides (NSP). The principal polysaccharides in cereal NSP are arabinoxylans (AX), cellulose and mixed linked (1- > 3; 1- > 4)-β-D-glucan (β-glucan), which, together with lignin, make up most of the cell wall and are referred to as dietary fiber (DF). The composition of the cell walls varies between the cellular tissues within the cereal grain and among similar tissues of different grains [6]. Barley has a husk layer that may remain even after the threshing process, whereas wheat husk is lost during the threshing of wheat, thus the DF content is approximately 50% higher in barley than in wheat [7]. These differences in DF content, which are counteracted by a higher starch concentration in wheat compared to barley [7], are responsible for the higher apparent total tract digestibility (ATTD) of wheat diets compared to barley diets in pigs [8].

Grinding is a physical process that reduces PS and increases the surface area, allowing better contact with digestive enzyme, which improves apparent ileal digestibility (AID) and ATTD [9]. This enables optimal nutrient utilization and enhances animal performance. Furthermore, particle size has been associated with changes in the microbial population, and coarse particles stimulate microbial fermentation of DF, which contributes to improved intestinal health by reducing ulceration and *E. coli* adhesion to the mucosa in the small intestine [10]. The hypothesis of the present study was that finely ground feed improves the digestibility of nutrients and the concentration of short-chain fatty acids (SCFA) in the digesta.

The objective of this study was to investigate the influence of PS of barley and wheat in diets on AID, ATTD, recovery of nutrients and DF, mean transit time (MTT) in the small and large intestines, and SCFA concentration and composition in growing pigs.

## 2. Materials and Methods

The study complied with the guidelines of the Danish Ministry of Justice, Act no. 474 of 15 May 2014, concerning experiments with animals and the care of experimental animals, as stipulated in the executive order no. 12 of 7 January 2016.

### 2.1. Diets

Four experimental diets that differed in cereal type (barley or wheat) and PS (fine or coarse) were used (Table 1). The grain components were ground for diets with a fine PS. To achieve approximately the same PS in the ground diets irrespective of grain type, it was found that barley should be ground using a 3 mm sieve and wheat should be ground using a 4.5 mm sieve in a hammer mill. The barley and wheat used in the coarse diets were rolled before inclusion. It was possible to produce rolled feed without the risk of whole grains in the feed. Chromic oxide (2 g/kg diet) was included in the diets as a marker for the determination of the AID and ATTD of nutrients and energy, and MTT.

### 2.2. Nimals and Experimental Designs

Two animal experiments were conducted to perform distinct analyses. The pigs (DanBred Genetics, Ballerup, Denmark) used in both experiments were from the pig herd of Aarhus University, Denmark.

### 2.3. Experiment 1

The experiment was conducted according to a 5 × 5 Latin square design, using five crossbred barrows [initial BW 35.9 ± 1.5 kg; (Danish Landrace × Yorkshire) × Duroc]. The pigs were fed five different diets, including the four experimental diets and a standard diet, over five periods, each with a duration of two weeks. However, the standard diet was not part of this study, and the results from the standard diet were not included in the statistical analyses. The pigs were fitted with a simple T-cannula at the ileum, approximately 15 cm anterior to the ileocecal junction, following previously outlined procedures [11]. The pigs were fed the same amount of daily net energy, and the amount of feed was adjusted throughout the experiment to match the body weight of the pigs. The feed was provided in three meals of equal size at 07:00, 15:00 and 22:00 h, and the meal size was gradually increased following feeding units for growing pigs [12]. Each experimental period consisted of 14 days: 8 days of adaptation to the experimental diets, followed by 3 days of feces collection and 3 days of ileal digesta collection. The pigs were placed in stainless steel metabolic crates on the last day of the adaptation period. Feces were collected from 07:00 to 15:00 on days 9 to 11. Digesta were collected from 07:00 to 15:00 on days 12 to 14. This approach has been shown to provide a representative sample of digesta that encompasses postprandial changes in nutrient flow [13]. During the collection period, digesta were collected every hour, weighed and immediately frozen (−20 °C). After 6 day collection, period the pigs were returned to their pens for 8 days of adaptation to the next experimental diet.

### 2.4. Experiment 2

This study involved 32 castrated male pigs with an initial weight of 30.8 ± 1.3 kg [(Danish Landrace × Yorkshire) × Duroc]. The experimental design was a randomized block design with eight blocks, each consisting of four pigs fed one of the four diets. The animals were fed equal amounts of feed on an energy basis at approximately 10% below *ad libitum* feed intake. The pigs were fed twice a day, and the meal size was gradually increased from 6.16 MJ to 7.70 MJ net energy per day as the animals grew [14]. The pigs were individually housed on a concrete floor with no bedding material.

The pigs were fed one of the four experimental diets for a period of four weeks, after which they were euthanized, and samples were collected. The animals were stunned followed by exsanguinations. The digestive tract was rapidly removed and divided into the following sections: the stomach, the small intestine, the cecum and four equal sections of the colon (Colon1, Colon2, Colon3 and Colon4). The total digesta of the small intestine, the cecum and the four colonic sections (Colon1–4) were collected and weighed. The samples were frozen and stored at −20 °C until needed for further analysis.

### 2.5. Chemical Analyses

All analyses were made in duplicate. Cr_2_O_3_, nitrogen and starch determinations were performed on wet material, while all other analyses were carried out on freeze-dried materials. SCFA was performed on freeze-dried material in Experiment 1 and on wet material in Experiment 2. The dry matter content of feed, digesta and feces was determined by drying at 105 °C until a constant weight was achieved. Protein (N × 6.25) was determined using the Kjeldahl method with a Kjell-Foss 16,200 autoanalyzer. Gross energy was determined using bomb calorimetry with a LECO AC 300 automated calorimeter system 789–500 (LECO, St Joseph, MI, USA). Fat was extracted with diethyl ether after acid hydrolysis [15]. Cr_2_O_3_ content was determined using the method described by Schurch et al. [16]. Digesta samples were analyzed for SCFA by gas chromatography as described in detail by Jensen et al. [17].

Sugars (glucose, fructose and sucrose) and fructans in feed, ileal digesta and fecal samples were analyzed using the enzymatic-colorimetric method of Larsson and Bengtsson [18], and the sucrose present as part of fructans was corrected as described by Bach Knudsen and Hessov [19]. Starch was analyzed using a modified enzymatic method as described by Bach Knudsen [7]. In feed, starch determination was also conducted without further milling preceding the analysis. In digesta and feces, starch was determined in wet and freeze-dried ground samples. Total β-glucan was determined using an enzymatic-colorimetric method [20]. Total non-starch polysaccharides (T-NSP) and their constituent sugars were determined as alditol acetates by gas-liquid chromatography for neutral sugars and by a colorimetric method for uronic acids, as described by Bach Knudsen [7]. Soluble NSP (S-SNP) in the starch-free residue was extracted using a phosphate buffer at neutral pH (0.2 mol/L, 100 °C, pH 7.0) [21], and the neutral and acidic sugars in insoluble NSP (I-NSP) were analyzed as previously described [7]. The content of cellulose was calculated as follows:Cellulose = NSP_glucose_ − β-glucan,
non-cellulosic polysaccharides (NCP) as:NCP = rhamnose + fucose + arabinose + xylose + mannose + galactose + (glucose- β-glucan) + uronic acids,
arabinoxylan (AX) as:AX = arabinose + xylose,
and S-NSP as:S-NSP = Total-NSP − I-NSP.

Klason lignin was measured gravimetrically as the residue resistant to 12 mol/L H_2_SO_4_ [22,23].

### 2.6. Calculations and Statistical Analyses

The apparent digestibility of nutrients at the terminal ileum and total tract were calculated relative to the indigestible marker (Cr_2_O_3_) content:$$\mathrm{Digestibility}\;\mathrm{of}\;X\;(\mathrm{of}\;\mathrm{intake})=[1-\frac{{\mathrm{Cr}}_{2}O_{3}\;diet\times X\;digesta/feces}{{\mathrm{Cr}}_{2}O_{3}\;digesta/feces\times X\;diet}]\;\times \;100$$
where X is the nutrient in question. X_(diet)_ and X_(digesta)_ are concentrations of specific nutrients in the diet and digesta from the terminal ileum or feces.

The quantitative flow (recovery) of nutrient X was calculated as follows:$$\mathrm{Flow}\;\mathrm{of}\;X\;(g/d)=[\frac{Intake\;of\;X\left(g/d\right)\times (100-digestibility\;of\;X)}{100}]$$
the mean transit time in the intestinal segments was calculated as follows:$$\mathrm{MTT}=\frac{{\mathrm{Cr}}_{2}O_{3(\mathrm{GI})}\times 24}{{\mathrm{Cr}}_{2}O_{3\mathrm{day}}}$$
where Cr_2_O_3(GI)_ and Cr_2_O_3day_ are the amounts of Cr_2_O_3_ in the specific GI segment and the daily intake of Cr_2_O_3._

Before the animal experiment, a power analysis was performed using SAS JMP based on previous experience with digestibility and SCFA concentration. Based on a power level of 80% and a significance level (α) of 0.05, the minimum required sample size for the study was determined to be 5 pigs. All data were analyzed as least squares means on the Fit Model platform of SAS JMP version 15. 0. 0 (SAS Inst. Inc., Cary, NC, USA). Statistical significance was determined at *p* < 0.05, and trends are considered for 0.05 ≤ *p* < 0.10. The least square means were calculated using a post-hoc Tukey test.

The data from Experiment 1 were analyzed as a Latin square design using two-way ANOVA:*Y_ijkl_* = *μ* + *p_i_* + *α**_j_* + *c_k_* + *s_l_* + *cs_kl_* + *ε**_ijkl_*,
where *Y_ijkl_* is the measured dependent variable, *μ* is the overall mean, *p_i_* is the random effect of period, *α*_j_ is the effect of animal, *c_k_* is the main effect of cereal types (*k* = barley and wheat), *p_l_* is the main effect of PS (*l* = fine or coarse), *cp_kl_* is the interaction between cereal types and PS, and *ε*_ijkl_ is the residual component.

The data from Experiment 2 were analyzed as a randomized block design using two-way ANOVA:*Y_jkl_* = *μ* + *b_j_* + *c_k_* + *s_l_* + *cs_kl_* + *ε**_jkl_*
where *Y* is the measured variable, μ is the overall mean, *b_j_* is the random effect of block, *c_k_* is the main effect of cereal types (*k* = barley or wheat), *s_l_* is the main effect of PS (*l* = fine or coarse), *cp_kl_* is the interaction between cereal types and PS, and *ε*_jkl_ is the residual component.

## 3. Results

### 3.1. Diets

The gross energy concentration of the diets was similar. However, a few differences between the diets, reflecting the differences between the cereal types and the PS of the feed, were observed (Table 1). The protein content was higher in the coarse diets when compared to their fine counterparts.

The starch analysis was performed both with and without milling preceding the analysis to evaluate how much starch was bound in the particles of the diets (Table 1). When the analysis was performed on the diets without milling, a difference between cereal type and PS of the feed was observed. The content of starch was highest in the fine wheat diet and lowest in the coarse barley diet. Milling the diets prior to analysis resulted in a higher starch content of the barley diets and the difference between the fine and coarse diet equaled out. In the wheat diets, the difference between the fine and coarse diets persisted after milling, and a higher starch content was observed for both diets when compared to the analysis done without milling.

The content of DF differed between the barley and wheat diets, being highest in the barley diets, whereas no difference due to PS was observed. The barley diets had a higher total and soluble NSP content than the wheat diets caused by a higher content of cellulose and β-glucan, whereas no difference was observed for AX and Klason lignin (Table 1).

Determination of the PS distribution showed that the coarse diets had the highest percentage of particles greater than 1 mm, and the fine diets were almost devoid of particles greater than 2 mm (Table 1). When comparing the coarse diets, the barley diet had the largest proportion of both particles greater than 1 and 2 mm.

### 3.2. Recovery of Ileal and Fecal Materials

The recovery of total ileal wet and solid digesta, organic matter (OM), total carbohydrates, nitrogen (g/d), fat (g/d), and residue (g/d) was higher when feeding the barley coarse diet compared with the other diets (Table 2; *p* < 0.05). In the recovery of fecal materials, the cereal and PS effects were interactive with respect to total solid fecal materials, OM, total carbohydrates (g/d), nitrogen (g/d), fat (g/d), and organic acids (g/d; *p* < 0.05). The recovery of nutrients in fecal material of pigs fed the barley coarse diet was higher than when feeding the wheat diets. PS and the type of cereal grain tended to have an interactive effect on fat (g/kg) solid, total solid in ileal material (g/d), and ash (g/d) in fecal material (0.05 ≤ *p* < 0.10).

### 3.3. Apparent Ileal and Total Tract Digestibility

Cereal and PS had an interactive effect on AID of most nutrients as well as DF and some of its components. The AID of OM, energy, starch, fat, total carbohydrates, NSP, cellulose, xylose and DF was lower in pigs fed the coarse barley diet compared to the fine barley diet and the wheat diets (*p* < 0.05; Table 3). No interaction between cereal and PS was observed for the AID of β-glucan, AX and arabinose (*p* > 0.05). Instead, the effect of PS was found for these components (*p* < 0.05), and an effect of cereal was seen for the AID of fructan (*p* = 0.027). Using different sample preparations prior to starch analyses showed that milling the diets and digesta prior to analyses resulted in a higher digestibility of starch at the ileal level for all diets compared to analyses done on the raw samples. The difference between the sample preparations was especially pronounced for the coarse barley diet, where the starch digestibility was increased from 89.1 to 92.2% at the terminal ileum (Table 3). Interactive tendencies were observed in the AID of fructans and AX, and ATTD of total carbohydrates (0.05 ≤ *p* < 0.10).

The apparent digestibilities of total NSP, cellulose, AX and β-glucan from the ileum to feces across different intestinal segments are shown in Figure 1. Overall, there was an interaction between cereal type and PS for the cecum and colon segments, with lower values found for the coarse barley diet compared to the other diets. There were also significant the degradation of NSP components along the large intestine; almost all β-glucan was degraded in the cecum, whereas cellulose was degraded at more distal locations, and with AX in-between. The negative digestibility values for cellulose in cecum are most likely caused by the separation of cellulose relative to the marker at this site in the small and large intestines during the pigs euthanization process. At the fecal level, a cereal effect was found for the ATTD of total carbohydrates, DF, NSP, AX, and cellulose (*p* < 0.05). The PS effect was observed for the ATTD of total carbohydrates, DF, NSP, AX and xylose. These differences also translated into lowest ATTD of OM, energy and fat for the pigs fed barley coarse diet and the highest ATTD values for the other diets (*p* < 0.05; Table 3).

### 3.4. The Relative Intestinal Weight, Digesta Weight, pH and Mean Transit Time in Digesta of the Small and Large Intestines

The experimental diets had no effect on the relative weight of the segments of the small and large intestines (Table 4). An interaction between cereal and PS was found for digesta weight in colon4 and for pH in colon3. The digesta weight in colon4 of pigs fed wheat coarse diet was higher than other diets, and barley and wheat fine diets were positioned between barley and wheat coarse diets. The pH in colon3 of pigs fed barley coarse diet was lower compared to other diets (Table 4; *p* < 0.05). A cereal effect was observed for digesta weight in colon1 and total colon, and pH in colon2 (*p* < 0.05), whereas no PS effect was found on digesta weight and pH of digesta.

Mean transit time in the cecum, colon segments, total colon, and overall (the small and large intestine) of pigs fed barley coarse diet was lower than for pigs fed barley fine or the wheat diets (Table 5; *p* < 0.05). No interaction was found for MTT in the small intestine (*p* > 0.05).

### 3.5. Concentration of Short Chain Fatty Acids in the Large Intestine

No interaction between cereal type and PS, and no PS effect on the concentration of SCFAs in digesta, was found (Table 6). Cereal type affected the total SCFA in the cecum, the proportion of acetic acid in the cecum, colon1, and colon2, and the proportion of branched-chain fatty acids (BCFA) in colon2 and colon3 (*p* < 0.05). Cereal tended to have an effect on the total SCFA in colon1 and colon2, the proportion of acetic acid in colon3, and the proportion of propionic acid in cecum (0.05 < *p* < 0.1).

## 4. Discussion

Grinding feed materials to reduce their PS is a conventional way to increase the surface area of the feed particles for improved nutrient digestibility and utilization. These aspects have been studied with different ingredients such as distillers’ dried grains with solubles, corn, soybean meal and soybean hulls. These studies have generally demonstrated improved AID and ATTD in growing pigs and improved feed efficiency without affecting gastric ulceration in growing pigs fed a corn-wheat-soybean meal-based diet [24]. However, PS distribution at low and high DF levels may influence gut health in different ways [25], and knowledge on how finely and coarsely ground European cereal feedstuffs such as wheat and barley with contrasting DF content influence AID and ATTD of nutrients and the degradation through the large intestine is lacking. In the current study, we found that pigs fed a coarse barley diet had lower AID and ATTD of nutrients compared with other diets. For the DF and its components, the ATTD of wheat and the fine PS diets were higher compared with coarse PS diets. The main reason for the difference in NSP and DF between barley and wheat is the presence of the husk layer in barley, which accounts for 10–15% of the whole grain [26]. In barley, the husk is tightly attached to the pericarp layer, whereas wheat loses its husk layer during threshing and therefore only has the pericarp and testa layers left as part of the grain [27,28]. Total wet and solid materials and other nutrients in ileal and fecal materials of pigs fed a barley coarse diet were higher compared with other diets. Furthermore, the dry matter content of the ileal digesta after feeding the coarse barley diet was higher, indicating that it was primarily the undigested residues induced by the coarse structure that caused the higher ileal digesta flow rather than differences in the physicochemical properties. The weight of digesta in the colon, however, was only influenced by cereal type. Furthermore, the MTT in pigs fed a barley coarse diet was lower than that of pigs fed other diets, suggesting that the digesta of barley coarse diet did not have enough time to be fermented in the large intestine. This phenomenon has previously been seen when the diet contained high insoluble fiber such as cellulose and insoluble NCP [29].

In the present study, the digestibility of NSP and its the main components—cellulose, AX and β-glucan—clearly increased during passage of the large intestine but at various rates according to the property of DF components and cereal type. The cellulose digestibility in ileum was not influenced by either cereal type or PS due to the insolubility of cellulose [30]. However, the digestibility of total NSP, AX and β-glucan was influenced by PS at the ileal level. β-glucan was already extensively degraded at the terminal ileum, as found in other studies with barley and oats [31], and almost completely degraded in the cecum, as also found with oats [32]. The degradation of cellulose and AX occurred more slowly and with a significant influence of PS for cellulose and for both cereal type and PS for AX. The degradation of AX was consistently lower for barley than for wheat, which is most likely caused by the structure of the AX in the husk layer and ferulic acid cross-linkages. The ferulic acid content, 731 µg/g in whole grain barley compared to 689 µg/g in whole grain wheat is known to hinder fermentation and degradation of the cell wall in the large intestine [33,34]. Generally, ferulic acid cross-linkages profoundly affect the degradation and fermentation of cell walls, decreasing digestibility in the small and large intestines [35].

The cereal type, PS, and level of DF did not influence the relative weight of the small intestine, cecum, and colon and digesta weight. The final body weight of the experimental pigs was approximately 52 ± 1.5 kg. Generally, smaller pigs have lower fermentation ability in the large intestine than larger pigs [36], and although there was a larger inflow of potentially fermentable carbohydrates to the large intestine with the barley diets, the total degradation of carbohydrates in the large intestine was only higher for the barley coarse diet. However, this had no influence on the relative weight of the large intestine. In contrast, in pigs exceeding a body weight of 100 kg, it has been shown that the fermentation of DF could lead to an increase in the weight of the small and large intestines [36,37].

The NSP content in the barley diets was higher compared to the wheat diets, which was expected to induce more fermentation in the large intestine, thereby increasing SCFA concentration. However, the SCFA concentration in pigs fed the wheat diets was higher than in pigs fed barley diets. This is probably related to the generally higher digesta weight in the colon of pigs fed barley diets [38]. The husk of barley has a rigid structure, and although the DF intake was higher for the barley diets compared to the wheat diets, it was only in the case of the barley coarse diet that the total degradation of carbohydrates in the large intestine was higher (245 g/d) compared to the other diets (151–158 g/d). In addition, the mean transit time, which was significantly lower for the barley coarse compared to the other diets, also seems to have limited importance for total SCFA in the large intestine. Unlike our results, Stewart and Slavin [38] reported that a finely ground aleurone by-product of wheat and small particle size of wheat bran showed higher SCFA concentrations in vitro compared to large particle size or coarsely grounded by-products probably due to increased accessible surface area. This difference may be caused by different microbial fermentation between wheat and barley diets, as barley β-glucan can decrease the abundance of *Bacteroides*, *Porphyromonas*, and *Prevotella* spp, which are related to DF fermentation [39]. In the current study, the β-glucan level in the barley diet (2.1–2.4 g/kg) is 4–5 times higher than in the wheat diets (0.4–0.5 g/kg), and thus there is a potential for higher β-glucan content in barley-based diets to impede the fermentation process with specific microbiota, thereby the PS effect became blurred.

## 5. Conclusions

In conclusion, the outcomes of our investigation revealed that the variation in PS within wheat-based diets did not significantly impact the digestibility and transit time of digesta within the small and large intestines. However, feeding a coarse barley diet resulted in substantially reduced digestibility and a faster MTT in comparison to both the fine barley diet and the wheat-based diets in growing pigs. This response is likely due to the structural difference between the barley hull and the outer layer of wheat. Therefore, when formulating diets for pigs, it is advisable to consider not only the PS but also the structural dissimilarities between the cereals being included.

## Figures and Tables

**Figure 1 animals-13-01986-f001:**
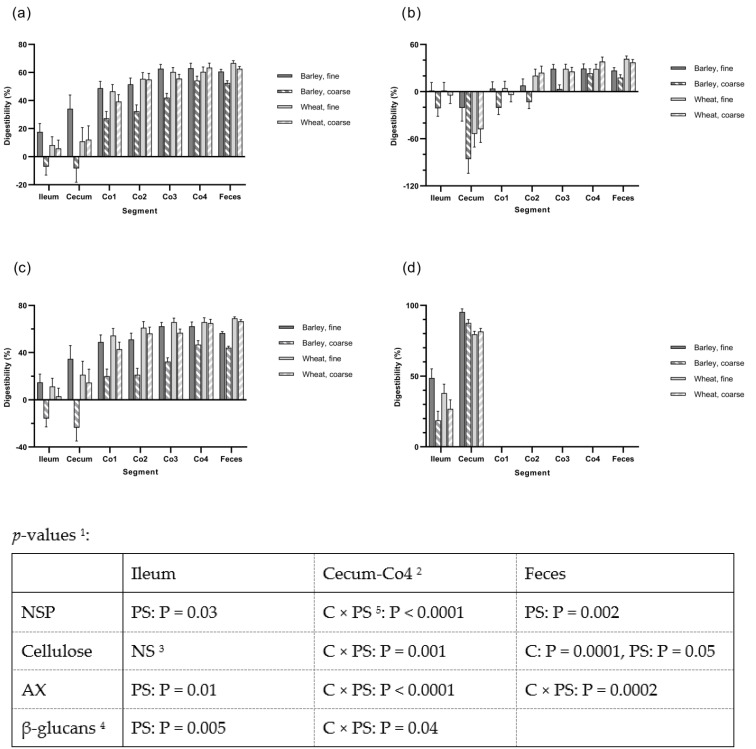
The digestibility of (**a**) Non-starch polysaccharides (NSP), (**b**) Cellulose, (**c**) Arabinoxylan (AX), and (**d**) β-glucan in ileal and fecal material from ileal cannulated pigs in Experiment 1(Experiment 1; *n* = 5) and in digesta from cecum and four segments of the colon from pigs slaughtered in experiment 2 (Experiment 2; *n* = 8). ^1^ PS: Particle size; C: Cereal. ^2^ The digestibilities in the segment’s cecum through Co4 were analyzed as repeated measurements with diet as the between-animal effect and segment as the within-animal effect. The effect of segment was significant in all cases (*p* < 0.0001) but no interactions between segment and particle size and cereal were observed. ^3^ NS: Not significant. ^4^ The content of β-glucan was only measured in ileal and cecal digesta. ^5^ C × P, the interaction between cereal type and particle size.

**Table 1 animals-13-01986-t001:** Ingredients and chemical composition of the experimental diets (g/kg dry matter).

	Diet			
	Barley	Barley	Wheat	Wheat
	Fine	Coarse	Fine	Coarse
Ingredient				
Wheat	-	-	699	699
Barley	699	699	-	-
Soybean meal (Toasted)	240	240	240	240
Animal fat	20	20	20	20
Molasses	10	10	10	10
Lysine 40%	3	3	3	3
Minerals and vitamins	26	26	26	26
Chemical composition				
Dry matter	913	917	911	915
Ash	6.1	7.4	5.7	6.9
Gross energy, MJ	18.6	18.7	18.6	18.4
Fat	55	57	48	51
Crude protein	221	244	216	234
Total carbohydrate	529.8	476.1	572.8	490.5
Sugars	2.9	3.0	2.4	3.0
Starch ^1^	374	312	449	365
Starch ^2^	434	439	494	468
Total NSP ^3^	152 (34)	160 (38)	120 (21)	121 (26)
Cellulose	6.8	8.0	5.7	6.4
Non-cellulosic polysaccharides	145	152	114	115
β-glucan	2.4	2.1	0.5	0.4
Arabinoxylan	6.0	5.9	5.9	5.4
Arabinose	2.4	2.4	2.5	2.3
Xylose	3.7	3.5	3.4	3.0
Fructan	0.9	1.1	1.4	1.5
Total non-digestible carbohydrate	153	161	121	123
Klason lignin	33	27	34	27
Dietary fiber	186	188	155	150
Particle size distribution, %				
>1 mm	16.3	51.9	24.1	37.8
>2 mm	0.7	23.3	2.9	12.0
Calculated energy content				
Net energy MJ/kg DM (FUgp) ^4^	9.39 (1.22)	9.39 (1.22)	10.00 (1.30)	10.00 (1.30)

^1^ Determined without further milling preceding the starch analysis, ^2^ Determined with milling preceding the starch analysis, ^3^ NSP, Non-starch polysaccharides, values in brackets are soluble NSP, ^4^ FUgp = Feed unit growing pigs.

**Table 2 animals-13-01986-t002:** Recovery of ileal and fecal materials of pigs fed the experimental diets (Experiment 1; *n* = 5).

	Barley		Wheat			*p*-Value		
	Fine	Coarse	Fine	Coarse	SEM ^1^	Cereal	Particle Size	C × P ^2^
Ileal materials								
Total wet materials, g/d	6071 ^b^	8604 ^a^	5520 ^b^	5864 ^b^	303	0.001	0.002	0.007
Total solid, g/d	524 ^b^	798 ^a^	475 ^b^	481 ^b^	24	0.001	0.001	0.001
Total solid, g/kg wet material	86 ^b^	95 ^a^	86 ^b^	83 ^b^	3	0.044	0.376	0.047
Ash, g/g	66 ^b^	99 ^a^	67 ^b^	66 ^b^	3	0.001	0.001	0.001
Organic matter, g/d	458 ^b^	699 ^a^	408 ^b^	415 ^b^	22	0.001	0.001	0.001
Total carbohydrates, g/d	257 ^b^	384 ^a^	229 ^b^	237 ^b^	14	0.001	0.002	0.003
Total carbohydrates, g/kg solid	486	486	485	493	13	0.804	0.773	0.749
Nitrogen × 6.25, g/d	69 ^b^	112 ^a^	66 ^b^	65 ^b^	4	0.001	0.001	0.001
Nitrogen × 6.25, g/kg solid	132 ^b^	140 ^a^	139 ^ab^	136 ^ab^	2	0.560	0.298	0.043
Fat, g/d	26 ^b^	43 ^a^	26 ^b^	28 ^b^	1	0.001	0.001	0.001
Fat, g/kg solid	49	55	55	57	1	0.001	0.001	0.070
Organic acids, g/d	26	40	24	22	4	0.054	0.207	0.102
Organic acids, g/kg solid	50	47	50	45	5	0.822	0.476	0.851
Residue, g/d	80 ^b^	119 ^a^	62 ^b^	62 ^b^	6	0.001	0.009	0.010
Fecal materials								
Total wet materials, g/d	1013	1396	879	1085	48	0.002	0.001	0.099
Total solid, g/d	288 ^b^	374 ^a^	238 ^c^	266 ^bc^	6	0.001	0.001	0.002
Total solid, g/kg wet material	284	268	270	246	8	0.057	0.037	0.630
Ash, g/g	49	59	47	53	1	0.013	0.001	0.078
Organic matter, g/d	240 ^b^	314 ^a^	191 ^d^	213 ^c^	7	0.001	0.001	0.004
Total carbohydrates, g/d	106 ^b^	139 ^a^	72 ^c^	79 ^c^	4	0.001	0.001	0.015
Total carbohydrates, g/kg solid	370	376	305	298	11	0.001	0.940	0.537
Nitrogen × 6.25, g/d	57 ^b^	80 ^a^	51 ^c^	60 ^b^	2	0.001	0.001	0.003
Nitrogen × 6.25, g/kg solid	200	214	214	226	5	0.040	0.037	0.830
Fat, g/d	29 ^c^	45 ^a^	30 ^c^	37 ^b^	1	0.008	0.001	0.006
Fat, g/kg solid	101	119	123	137	4	0.001	0.020	0.649
Organic acids, g/d	9 ^bc^	13 ^a^	8 ^c^	9 ^b^	0.3	0.001	0.001	0.001
Organic acids, g/kg solid	30	35	34	35	1	0.109	0.011	0.143
Residue, g/d	38	37	30	28	4	0.093	0.689	0.858

^1^ SEM, standard error of mean, ^2^ C × P, the interaction between cereal type and particle size, ^a–c^ Row with different superscript letters is significantly different (*p* < 0.05).

**Table 3 animals-13-01986-t003:** Apparent digestibility of nutrients and fiber at the terminal ileum and feces in pigs fed experimental diets (Experiment 1; *n* = 5).

	Diet							
	Barley		Wheat			*p*-Value		
	Fine	Coarse	Fine	Coarse	SEM ^4^	Cereal	Particle Size	C × P ^5^
Apparent ileal digestibility								
Organic matter	73.8 ^a^	60.3 ^b^	75.7 ^a^	75.2 ^a^	0.95	0.001	0.001	0.001
Energy	73.6 ^a^	60.1 ^b^	75.0 ^a^	74.6 ^a^	0.98	0.001	0.001	0.001
Ash	41.8 ^ab^	29.6 ^b^	33.7 ^b^	46.6 ^a^	2.09	0.065	0.863	0.001
Fat	74.7 ^a^	59.6 ^c^	69.6 ^b^	69.9 ^b^	1.23	0.069	0.001	0.001
Total carbohydrates	77.9 ^a^	68.3 ^b^	80.2 ^a^	79.1 ^a^	1.18	0.001	0.002	0.007
Sugars	76.4	75.0	74.1	72.6	3.79	0.553	0.709	0.995
Fructan	97.9	81.3	79.2	79.1	4.21	0.027	0.078	0.092
Starch ^1^	95.0 ^a^	89.1 ^b^	95.9 ^a^	94.8 ^a^	0.91	0.007	0.005	0.028
Starch ^2^	96.5 ^a^	92.2 ^b^	96.3 ^a^	96.3 ^a^	0.47	0.003	0.002	0.002
Dietary fiber	26.3 ^a^	−1.18 ^b^	24.1 ^a^	18.2 ^a^	3.68	0.048	0.002	0.019
Total NSP ^3^	22.3 ^a^	−1.5 ^b^	11.8 ^ab^	8.3 ^ab^	4.18	0.909	0.011	0.040
β-glucan	52.8	24.7	42.6	32.6	6.31	0.657	0.013	0.214
Cellulose	14.7 ^a^	−2.0 ^b^	9.6 ^ab^	6.2 ^b^	4.12	0.709	0.144	0.041
Arabinoxylan	19.9	−9.5	14.7	5.3	4.46	0.312	0.002	0.055
Arabinose	26.2	6.3	20.8	13.4	3.90	0.834	0.008	0.148
Xylose	15.8 ^a^	−20.7 ^b^	10.2 ^a^	−1.0 ^ab^	4.96	0.194	0.001	0.035
Apparent total tract digestibility								
Organic matter	86.3 ^b^	82.1 ^c^	88.5 ^a^	87.2 ^b^	0.35	0.001	0.001	0.003
Energy	84.3 ^b^	79.1 ^c^	86.2 ^a^	84.4 ^b^	0.39	0.001	0.001	0.003
Ash	57.9	57.5	53.8	57.5	1.24	0.143	0.213	0.129
Fat	71.7 ^a^	58.8 ^c^	65.5 ^b^	60.1 ^c^	1.03	0.049	0.001	0.006
Total carbohydrates	90.9	88.4	93.7	93.0	0.39	0.001	0.004	0.058
Dietary fiber	58.2	47.7	64.8	57.7	1.69	0.001	0.001	0.333
Total NSP	62.6	54.8	67.8	63.3	1.73	0.004	0.008	0.361
Cellulose	57.2	47.2	52.3	45.1	2.75	0.239	0.014	0.636
Arabinoxylan	58.7	46.9	70.0	64.1	1.93	0.001	0.002	0.165
Arabinose	74.0	69.1	67.0	63.2	2.06	0.014	0.070	0.792
Xylose	48.8 ^c^	31.2 ^d^	72.2 ^a^	64.7 ^b^	2.18	0.001	0.001	0.049

^1^ Starch in the diet determined without further milling preceding the analysis and starch in digesta determined in wet material. ^2^ Starch in the diet determined after milling the sample and starch in digesta determined in freeze-dried and ground material. ^3^ NSP, Non-starch polysaccharides, ^4^ SEM, standard error of mean, ^5^ C × P, the interaction between cereal type and particle size. ^a–d^ Row with different superscript letters is significantly different (*p* < 0.05).

**Table 4 animals-13-01986-t004:** The relative weight, digesta weight and pH in different intestinal segments of pigs fed experimental diets (Experiment 2; *n* = 8).

	Segment	Barley		Wheat			*p*-Value		
Item		Fine	Coarse	Fine	Coarse	SEM ^2^	Cereal	Particle Size	C × P ^3^
Relative weight ^4^	SI ^1^, %	4.2	4.4	4.6	4.5	0.37	0.456	0.783	0.636
	Cecum, %	0.6	0.5	0.5	0.5	0.05	0.608	0.637	0.846
	Colon1, %	1.6	1.7	1.9	2.4	0.41	0.282	0.518	0.626
	Colon2, %	0.9	0.8	0.8	0.7	0.12	0.432	0.322	0.924
	Colon3, %	1.1	0.9	0.9	0.9	0.09	0.239	0.307	0.460
	Colon4, %	1.1	1.0	1.2	1.2	0.07	0.050	0.794	0.367
	Total colon, %	4.2	4.7	4.7	4.5	0.28	0.606	0.616	0.236
Digesta weight	SI, g	267	218	206	266	32	0.843	0.867	0.098
	Cecum, g	314	302	351	302	43	0.667	0.480	0.677
	Colon1, g	580	485	368	431	44	0.006	0.721	0.084
	Colon2, g	371	416	383	357	42	0.577	0.834	0.401
	Colon3, g	292	337	249	246	36	0.076	0.568	0.528
	Colon4, g	243 ^ab^	156 ^b^	173 ^ab^	349 ^a^	32	0.721	0.868	0.018
	Total colon, g	1486	1393	1173	1283	92	0.030	0.926	0.281
pH	SI	5.7	6.1	5.8	5.7	0.19	0.353	0.412	0.169
	Cecum	5.4	5.3	5.3	5.4	0.07	0.161	0.382	0.069
	Colon1	5.5	5.4	5.5	5.6	0.09	0.341	0.625	0.536
	Colon2	5.8	5.7	5.8	6.2	0.10	0.035	0.186	0.053
	Colon3	6.3 ^a^	5.9 ^b^	6.3 ^a^	6.4 ^a^	0.12	0.073	0.258	0.028
	Colon4	6.5	6.3	6.4	6.4	0.07	0.837	0.904	0.128
	Average colon	6.0 ^ab^	5.9 ^b^	6.0 ^ab^	6.1 ^a^	0.07	0.088	0.868	0.036

^1^ SI, small intestine, ^2^ SEM, standard error of mean, ^3^ C × P, the interaction between cereal type and particle size. ^4^ Relative weight in the small and large intestines to body weight (%) = organ weight (kg)/body weight of pig (kg) × 100. ^a,b^ Row with different superscript letters is significantly different (*p* < 0.05).

**Table 5 animals-13-01986-t005:** Mean transit time of digesta in the small and large intestines of pigs fed the experimental diets (Experiment 2; *n* = 8).

	Diet							
	Barley		Wheat			*p*-Value		
	Fine	Coarse	Fine	Coarse	SEM ^1^	Cereal	Particle Size	C × P ^2^
Small intestine	4.8	4.1	5.3	4.4	0.43	0.431	0.097	0.872
Cecum	4.4 ^a^	2.6 ^b^	3.7 ^a^	4.2 ^a^	0.24	0.072	0.010	0.001
Colon1	5.5 ^a^	3.7 ^b^	5.7 ^a^	5.5 ^a^	0.34	0.007	0.008	0.034
Colon2	6.0 ^b^	4.0 ^c^	6.8 ^ab^	7.0 ^a^	0.31	0.001	0.006	0.001
Colon3	7.0 ^a^	4.4 ^b^	7.9 ^a^	7.0 ^a^	0.34	0.001	0.001	0.015
Colon4	7.4 ^a^	5.3 ^b^	7.9 ^a^	7.7 ^a^	0.42	0.002	0.015	0.046
Total colon	25.9 ^a^	17.4 ^b^	28.3 ^a^	27.2 ^a^	1.21	0.001	0.001	0.004
Overall	35.0 ^a^	24.1 ^b^	35.7 ^a^	36.3 ^a^	1.34	0.001	0.001	0.001

^1^ SEM, standard error of mean, ^2^ C × P, the interaction between cereal type and particle size. ^a–c^ Row with different superscript letters is significantly different (*p* < 0.05).

**Table 6 animals-13-01986-t006:** The concentration of short chain fatty acids in digesta of the small and large intestines in growing pigs (Experiment 2; *n* = 8).

	Barley		Wheat			*p*-Value		
	Fine	Coarse	Fine	Coarse	SEM ^3^	Cereal	Particle Size	C × P ^4^
Total SCFA ^1^, mmol/kg								
Small intestine	17	20	25	19	3.8	0.380	0.628	0.238
Cecum	123	135	177	192	26.7	0.046	0.625	0.957
Colon1	138	163	184	216	24.0	0.055	0.245	0.895
Colon2	143	137	189	201	27.0	0.052	0.913	0.749
Colon3	112	131	160	155	23.4	0.133	0.746	0.611
Colon4	118	123	157	153	23.2	0.144	0.979	0.850
Proportion of acetic acid, %								
Small intestine	71	80	76	79	2.8	0.233	0.989	0.191
Cecum	56	54	56	58	15.5	0.035	0.616	0.816
Colon1	52	53	57	56	13.9	0.025	0.312	0.959
Colon2	52	53	57	57	14.6	0.015	0.890	0.766
Colon3	57	53	58	59	14.7	0.087	0.924	0.796
Colon4	58	54	60	57	14.2	0.127	0.755	0.843
Proportion of propionic acid, %								
Small intestine	15	7	14	6	1.3	0.765	0.207	0.688
Cecum	26	30	29	28	8.1	0.066	0.499	0.705
Colon1	27	29	26	28	7.2	0.120	0.121	0.946
Colon2	25	28	24	25	7.9	0.161	0.704	0.865
Colon3	23	26	23	25	6.3	0.205	0.486	0.599
Colon4	22	24	22	23	2.1	0.186	0.644	0.836
Proportion of butyric acid, %								
Small intestine	14	13	8	9	0.7	0.478	0.931	0.716
Cecum	15	14	14	13	3.1	0.102	0.932	0.934
Colon1	17	15	14	14	3.4	0.238	0.433	0.748
Colon2	18	17	15	15	4.0	0.188	0.768	0.679
Colon3	15	18	15	14	2.8	0.492	0.602	0.089
Colon4	15	17	15	16	3.3	0.169	0.518	0.678
Proportion of valeric acid, %								
Small intestine	1	1	1	1	0.6	0.543	0.351	0.442
Cecum	3	2	2	2	0.6	0.478	0.961	0.268
Colon1	4	3	3	3	0.9	0.803	0.645	0.306
Colon2	5	3	3	3	1.2	0.853	0.462	0.289
Colon3	4	4	3	3	0.8	0.576	0.681	0.961
Colon4	4	4	3	4	1.0	0.401	0.599	0.415
Proportion of BCFA ^2^, %								
Small intestine	8	8	5	2	0.6	0.348	0.635	0.378
Cecum	1	1	1	1	0.4	0.310	0.634	0.317
Colon1	2	1	1	1	0.4	0.846	0.724	0.427
Colon2	2	1	2	2	0.4	0.002	0.772	0.116
Colon3	4	2	3	3	0.6	0.014	0.289	0.453
Colon4	5	4	4	5	1.2	0.211	0.773	0.283

^1^ SCFA, Short-chain fatty acid, ^2^ BCFA, Branched-chain fatty acid, ^3^ SEM, standard error of mean, ^4^ C × P, the interaction between cereal type and particle size.

## Data Availability

Not applicable.

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
