# Peer review of "Effect of Particle Size of Wheat and Barley Grain on the Digestibility and Fermentation of Carbohydrates in the Small and Large Intestines of Growing Pigs"

_animals, 2023, doi:10.3390/ani13121986_

Round 1

Reviewer 1 Report (Previous Reviewer 3)

I have no more comments. The study is worth to be published now.

Author Response

Dear Reviewer

We express our utmost gratitude for your invaluable assessment of our research endeavor. It is with sincere appreciation that we receive your positive evaluation. As we enter the summer season, we wish you a delightful and fulfilling time ahead.

With great respect,

Geonil Lee

Reviewer 2 Report (Previous Reviewer 4)

I have to state that the manuscript has not been substantially improved after the revision. Still errors concerning basic anatomy are present. It is clear that not enough attention has been done to the revision.

Line 31 – How the authors were able to weight the whole gastrointestinal tract (from mouth to anus)?

Line 102 – anatomically “terminal ileum” does not exist.

Line 126 – From anatomical point of view gastrointestinal tract (also known as alimentary tract or digestive tract) is a tract which passes food from mouth to the anus. I have the impression that the authors limit the GIT tract to segments located distally from the stomach.

Line 127 – which precisely part of the small intestine was dissected out?

Line 128 – It is not clear why the authors decided to study the digesta of ileum (but not from the duodenum or jejunum). It must be carefully explained.

Line 179, 186 – what post-hoc test was used for ANOVA?

Line 235 – ileum is a part of GIT. So “ileal and gastrointestinal” sounds awkward.

Author Response

Dear Reviewer

We extend our sincere gratitude for your invaluable feedback on our manuscript. Your insightful comments and advice have been instrumental in enhancing the overall quality of our paper.

In order to address your suggestions and incorporate necessary revisions, we have prepared a response letter detailing the specific changes made. Kindly find the attached Word file containing the response letter for your perusal.

Thank you once again for your invaluable contribution to our research.

Kind regards,

Geonil Lee

Reviewer 3 Report (New Reviewer)

Affiliation: I do not understand why the first author's affiliation is determined by 2 and the others by 1. It is more logical that the order of the affiliations is correlated to the order of authors

Simple summary (L17): and modulate of nutrient digestibility

Abstract (L25): In this section I miss a brief introduction to the topic

Introduction: 

L45 instead of single stomached use monogastric

L48:it repeats the information given in lines 43-44, so I suggest to eliminate it

L51: can have a negative impact on particularly gastric ulcers: I do not understand this sentence. Do you mean particularly as principally or as severe?

L52 I would erase "cereals in the form of"

L67: enhanced

L70 E coli should be written in italics

L81 and lines 87 and 88 If adding "only" in L81, I think Line 87 ("the PS distribution...) to l88 ((Table1)) can be deleted and language is more concise. 

Material and methods

Please revise table and figures formats to follow the journal's specifications. 

L95: If pigs in both experiments were similar, I would explain breed and age in this line. I am not sure of this because animals in experiment 1 are well described, but are not clear in experiment 2 (You don not mention the breed used)

L104: The amount of feed

L106: 07:00, 15:00, 22:00 hours

L110 and 111: 07:00 to 15:00

L128 and 129: I would write in the bracket of L128 Colon1, Colon2, Colon3 and Colon4 and in the bracket of L129 C1-4

L131 (chemicals analysis): In the results, you talk about BCFA, but no explanation on how they are determined is given in the text

L146 and L151:By Bach Knudsen and Knud Erik

L193-19: I would write this sentence (until [31]) before talking about the ANOVAs used in each experiment. Also, the point should be written after the citation. 

Results

Please explain in each table the meaning of SEM and CxP

L209: with and without

L214: delete pronounced

L300: No interaction

L 314-315: If the citation refers to both ideas Improved feed efficiency and no gastric ulceration, I would put it at the end of se sentence. If not, please provide the citation for the gastric ulceration

L317: I think it would be easier for the reader to have a .Knowledge... after citation 26

L226 (paragraph): Table 2 should be written after being metioned in the text. 

L234 fat (gkg) AND solid

L253: there were interactions

L264: wheat finE diet. Also, I would delete from and with the two other diet... until the first bracket

Figure 1: The description of the figure should go under the figure. I think it would be easier for the reader to have the name of the ingredient above each graph

Minor corrections shoud be done

Author Response

Dear Reviewer

We extend our sincere gratitude for your invaluable feedback on our manuscript. Your insightful comments and advice have been instrumental in enhancing the overall quality of our paper.

In order to address your suggestions and incorporate necessary revisions, we have prepared a response letter detailing the specific changes made. Kindly find the attached Word file containing the response letter for your perusal.

Thank you once again for your invaluable contribution to our research.

Kind regards,

Geonil Lee

Reviewer 4 Report (New Reviewer)

L18 – Insert  pig…intestinal tract.

L22 – Other diets?  What is diets?

L355 – 52 Kg. Insert SD in measure. And include (data not shown)

Quality of English is good. 

Author Response

Dear Reviewer

We extend our sincere gratitude for your invaluable feedback on our manuscript. Your insightful comments and advice have been instrumental in enhancing the overall quality of our paper.

In order to address your suggestions and incorporate necessary revisions, we have prepared a response letter detailing the specific changes made. Kindly find the attached Word file containing the response letter for your perusal.

Thank you once again for your invaluable contribution to our research.

Kind regards,

Geonil Lee

Reviewer 5 Report (New Reviewer)

This study investigated the effects of particle size and cereal type on the digestibility and fermentation of carbohydrates in the gastrointestinal tract of growing pigs. The authors carried out two experiments: in experiment 1, five pigs were fed five different diets over five periods, each with a duration of two weeks; in experiment 2, a total of 32 pigs were fed with four deferent diets with a duration of four weeks. The authors analyzed the digestibility, the relative weight, digesta weight and pH, mean transit time of digesta in the small and large intestines, and the concentration of short chain fatty acids in digesta of intestinal tracts. However, the author did not report on the weight gain of experimental growing pigs, which is a very important economic indicator. No matter how good the indicators measured in the laboratory were, if they contradict the weight gain data, they would not work. Therefore, if there were data on weight gain and feed conversion rate of growing pigs in Experiment 2, it is recommended that the author supplement them.

This manuscript had been reviewed by two reviewers, and the author had made revisions based on their suggestions, resulting in a significant improvement in the quality of the manuscript. Therefore, I only have a few comments:

Line 98, crossbred barrows (initial BW 35.91.5 kg; Danish Landrace ×Yorkshire × Duroc), Is Duroc the terminal paternal? Please mark it out.

Line 186-199, What statistical models or methods were used that were not specified, and what were the fixed and random effects in the model? A two-way ANOVA is not good for this study. A mixed model should be used. Because the block should be a random effect. Additionally, should the initial weight of pigs be considered as a covariate?

Figure 1, The digestibility of a) Non-starch polysaccharides (NSP), b) cellulose, c) Arabinoxylans (AX) in ileal and cecum can be a negative value? I can't understand. 

line 32, “lowercompared to” should be “lower compared to”

Line 213: “pronounced higher” should be changed to “pronouncedly higher” or “significantly higher”

Line 263: “wheat find diet” should be changed to “wheat fine diet”

Author Response

Dear Reviewer

We extend our sincere gratitude for your invaluable feedback on our manuscript. Your insightful comments and advice have been instrumental in enhancing the overall quality of our paper.

In order to address your suggestions and incorporate necessary revisions, we have prepared a response letter detailing the specific changes made. Kindly find the attached Word file containing the response letter for your perusal.

Thank you once again for your invaluable contribution to our research.

Kind regards,

Geonil Lee

Round 2

Reviewer 2 Report (Previous Reviewer 4)

Line 3, 20 and the rest of the manuscript - The authors must follow Nomina Anatomica Veterinaria. The term "intestinal tract" does not appear in NAV. Please use the correct anatomical names (I suggest just small intestine and large intestine).

Line 188 and 195 - the authors wrote: "The data from experiment 1 was analyzed as Latin square design using two-way ANOVA" and "The data from experiment 2 were analyzed as a randomized block design using two-way ANOVA". So, my question still remains unanswered.

Author Response

Dear Reviewer

We greatly appreciate your thoughtful review of our manuscript. In response to your valuable suggestions, we have made revisions to the content.

We have attached the response letter as word file. 

Kind regards,

Geonil Lee

This manuscript is a resubmission of an earlier submission. The following is a list of the peer review reports and author responses from that submission.

Round 1

Reviewer 1 Report

·         This manuscript has investigated the effect of barley and wheat with different particle sizes on the digestibility and fermentation of carbohydrates in growing pigs. The topic is important to demonstrate the particle size effects on nutrient digestibility associated with dietary fiber properties in barley and wheat. However, this manuscript needs a significant revision before it can be considered as a publication in this journal.

·         Comments:

·         Please add chemical composition (analyzed) of barley and wheat ingredients. Please state that this study used the same ingredients and ground them to different particle sizes. Also need to explain how the protein content could be different between particle sizes.

·         L103 stated that the pigs were fed the same amount of daily net energy. Please add net energy content of each diet (calculated) and how much were fed to pigs.

·         Table 1, what do the values in parentheses for Total NSP mean?

·         L126, total content of small intestine?

·         L142, please check if Bach is included in the citation as the last name.

·         L150, NSP, not SNP

·         L160, S-NSP was measured but no values are in any of tables

·         L181-191, please check the Pl, CPkl, CSkl, etc. in the statistical model equations.

·         Please add statement about further statistical analysis when an interaction was detected.- One-way ANOVA?

·         Also, this study declared trends with P<0.10 but had no statements regarding trends observed in many measurements. Please add them.

·         L201, must be Table 1

·         If this is calculated from digestibility with daily intake, I think AID and ATTD of these nutrients are better options in Table 2 instead of the fecal recovery since this did not use total collection method but just calculated the daily fecal recovery by digestibility from indirect method.

·         Please indicate Experiment number (1 or 2) in each table.

·         L210, protein content is about 10% different between fine and coarse diets.

·         L216, there is no soluble NSP content in the table.

·         This study used a factorial design with main effects of cereal types and particle sizes. However, the results did not state any of main effects but just interactions. As barley and wheat made differences in fiber composition, and particle sizes also made differences in digestibility, these the results of main effects need to be stated in the Results and discussed. This manuscript only stated one-way ANOVA results as there were interactions between these 2 factors. Please add the main effect results and discussion in the manuscript.

·         L246-254, Since there are no data for this figure 1, it is hard to understand the statements in these lines. Also, The p-values in Figure 1 may need to correspond to Table 3 since both are about digestibility (ileum and feces) but it didn’t. More clarification and data for this figure are needed. Cecum-Co4 means hindgut disappearance?

·         L255, cellulose is not significant for the cereal effect.

·         L277 needs more specific information such as it was higher/lower in barley than wheat.

·         L273, although this manuscript declared the p-value for trends, result statements missed that. Mainly in Table 4.

·         Also, the result statements for main effects (cereals/particle size) are missing.

·         There is no Table 6 for short chain fatty acids. Table 5 is repeated in L296. So short chain fatty acid data can’t be reviewed.

·         The discussion is not complete about particle size and cereal effects.

·         L302 and L305 need reference.

·         L316 mentioned that barley contains husk layers but needs more discussion about how this husk alter chemical composition of barley and digestibility of fiber components.

·         L320, since the recovery was calculated from digestibility, it has to be mirrored.

·         L347, needs more discussion about what no effect of treatments in relative weight means.

·         L353-L372 can’t be reviewed.

Not for quality of English but presentation may need to be improved along with putting correct information. 

Reviewer 2 Report

The manuscript of Lee et al. mainly explored the effects of cereal type (barley and wheat) with different particle size on digestibility and gut fermentation of growing pigs. The manuscript provides a good introduction to the subject, and the authors have used an intense methodology to demonstrate the conclusions they reach in the study. As the manuscript is well-written and easy to follow, I have only minor comments that need to be addressed:

L (line) 68-69: How coarse particles improve intestinal health? Please add some examples.

L 69-70: You mentioned that coarse particles stimulate microbial fermentation (L68). Maybe I have missed something, but your hypothesis is opposite to that. 

L 132-133: More details about sample preparation to quantify SCFA concentration in digesta are needed. Was metaphosphoric acid added to digesta samples? The table 5 is duplicated. The table 6 is missing and should be included in the results section. 

L 309-315: This paragraph is a mere repetition of the results; thus, I strongly recommend that you rewrite it.

Reviewer 3 Report

Comments to the Authors of manuscript number: animals-2348469 entitled “Effect of particle size of wheat and barley grain on the digestibility and fermentation of carbohydrates in the gastrointestinal tract of growing pigs”.

1. L 18- it is a gastrointestinal tract and not plural

2. L 26 what is recovery of ileal digesta?

3. L 96 – what was  the goal of the use of Latin square

4. Experiment 1 is not fully understood. Was there one pig subjected to one diet? What can show the results received from one animal? Were pigs fed 3 time a day? In general, pigs are fed two times.

5. L 178 – what was the power test for the experiment 1? It should be provided, especially that it seems there was 1 pig in the one group. The table 2 includes all these data which were obtained from this part. It should be clarify

6. L 186 – what was the power test for these analyses. There was only 4 pigs in one group. It should be presented.

7. Table 3. How is it possible n=5? It should be explained

8. it should be explained what parameter was analyzed and how many animals were taken into account. The 1 experiment included 4 pigs, and the 2 experiment included 32 (8 blocks per 4 pigs). It should be explained how pigs were distributed.

9. where is figure 1?

10. table in the figure 1 - How was calculated the digestibility in experiment 1, if total amount of pigs was 5 and 5 diets?

11. table 4 – how relative weight was calculated?

12. in the part of methods it should be described how all segments of intestine were distinguish

13. L 289- the analysis of fatty acids was not mentioned in material and methods

Reviewer 4 Report

The manuscript is nicely done and written. The study design is appropriate and apparently, the analyses were carefully performed.  I believe that the results are valuable for the scientific community and has significant scientific merit, as it will probably ignite many further studies in the near future.

However, some points need to be clarified before the publication.

Line 95 – it is unclear how many animals were used for experiment 1.

Line 101 – anatomically “terminal ileum” does not exist.

Line 118 – please write “ad libitum” in italics

Line 120 – please change the reference format to numerical

Line 124 - From anatomical point of view gastrointestinal tract (also known as alimentary tract or digestive tract) is a tract which passes food from mouth to the anus. I have the impression that the authors limit the GIT tract to segments located distally from the stomach.

Line 125 – which precisely part of the small intestine was dissected out?

Line 132 – the first appearance of SCFA is in line 70 and this is the place where it should be abbreviated